# Influence of trip distance and population density on intra-city mobility patterns in Tokyo during COVID-19 pandemic

**Kazufumi Tsuboi**[1]*, **Naoya Fujiwara**[1,2,3,4,5], **Ryo Itoh**[1]

1 Graduate School of Information Sciences, Tohoku University, Sendai, Miyagi, Japan, 2 Institute of Industrial Science, The University of Tokyo, Meguro, Tokyo, Japan, 3 Center for Spatial Information Science, The University of Tokyo, Kashiwa, Chiba, Japan, 4 Service Research Division, Tough Cyberphysical AI Research Center, Tohoku University, Sendai, Miyagi, Japan, 5 PRESTO, Japan Science and Technology Agency, Kawaguchi, Saitama, Japan

☯ These authors contributed equally to this work.
* tsuboi@se.is.tohoku.ac.jp

**Data Availability Statement:** The data used in this study is named mobile spatial statistics data, which is a third-party data that is collected, processed, and provided by DOCOMO Insight Marketing, INQ. The detailed information of the dataset is as

## Abstract

This study investigates the influence of infection cases of COVID-19 and two non-compulsory lockdowns on human mobility within the Tokyo metropolitan area. Using the data of hourly staying population in each 500m×500m cell and their city-level residency, we show that long-distance trips or trips to crowded places decrease significantly when infection cases increase. The same result holds for the two lockdowns, although the second lockdown was less effective. Hence, Japanese non-compulsory lockdowns influence mobility in a similar way to the increase in infection cases. This means that they are accepted as alarm triggers for people who are at risk of contracting COVID-19.

## Introduction

Allegedly originating from Wuhan, Hubei province, China, the novel coronavirus disease (COVID-19) has prevailed all over the world to infect more than 200 million people and killed more than 4 million people as of August 2021 [1]. The pandemic has changed our daily lives drastically. We maintain social distance and wear masks to save ourselves from infection. People stay home, work online; all these changes significantly reduce our daily mobility within cities.

Governments in many countries have also conducted non-pharmaceutical interventions (NPIs), including lockdown policies. NPIs have ordered or asked people to stay at home, encouraged onsite workers to start teleworking, and closed schools and public facilities for several weeks [2]. The type of NPIs vary across countries; most countries and cities imposed compulsory lockdowns with penalties for non-compliance, while some countries and cities, such as Sweden, Japan, and New York, imposed almost non-compulsory ones as summarized in [3]. Compulsory lockdowns are usually known to be more effective in mitigating infections than non-compulsory lockdowns [4]. However, there are concerns that long and strong restrictions cause a significant loss of income and opportunities, which can decrease people's welfare [5,

follows: (https://www.nttdocomo.co.jp/corporate/disclosure/mobile_spatial_statistics/). The authors did not have any special privileges in accessing the data and that other researchers would not have. Data access can be requested by obtaining a license contract with DOCOMO Insight Marketing, INQ. The authors are legally bound by the inclusive license contract between DOCOMO Insight Marketing, INQ. and Tohoku University, not to share as open access the detailed population data we analyze in this manuscript. The license contract puts the legal restriction in transferring the license. The authors need permission by DOCOMO Insight Marketing, INQ. when they publish their research outputs using the mobile spatial statistics data. Any parties wishing to access the mobile spatial statistics data may do so by contacting DOCOMO Insight Marketing, INQ. Address: Hareza Tower 17F, 1-18-1, Higashi Ikebukuro, Toshima-ku, Tokyo, Japan. URL: https://mobaku.jp/ Email: mobaku-ml@dcm-im.com.

**Funding:** This work was supported by the Research Institute for Mathematical Sciences, International Joint Usage/Research Center located at Kyoto University and the Open-type Professional Development Program (2021-SHIKOIN-7007). K.T. was supported by JST SPRING, Grant Number JPMJSP2114. N.F. was supported by JSPS KAKENHI Grant Number JP18K11462 and Promoting Grants for Research Toward Resilient Society, 2021. R.I. was supported by JSPS KAKENHI Grant Number JP18K01561. The funders had no role in study design, data collection and analysis, decision to publish, or preparation of the manuscript.

**Competing interests:** The authors have declared that no competing interests exist.

6]. When the pandemic lasts a long time, we need to consider less restrictive political options such as non-compulsory lockdowns to mitigate infection while keeping the damage to the economy small. For that purpose, we shed more light on people's decision making through which such moderate restrictions matter for daily mobility.

Many previous studies have examined the relationship between human mobility during COVID-19 and spread of the disease, which is promoted by the recent growing accessibility to mobile phone data. Many studies showed that the lockdowns have negative impact on mobility, and mitigate the spread of the disease [7–9]. Furthermore, some studies focused on detailed attributes of mobility to investigate how infection risks and lockdowns affect mobility through decision making, motivated by similar interests to ours. Focusing on travel distance of inter-county trips, it was shown that lockdowns not only reduce mobility, but also cause structural changes in mobility, such that many people quit long-distance travel and their networks become smaller [10, 11]. Other studies also show that effects of infection risks and lockdowns differ by objective of trip and attributes of travellers (e.g. age, gender, and occupation) in inter-city and itra-city mobility [11–15].

The differences between compulsory and non-compulsory lockdowns have also been examined. Ref [4] compared various countries' lockdowns and revealed that countries conducting compulsory lockdowns reduce infection cases and recover more rapidly than non-compulsory lockdowns such as in the United States and Sweden. They claimed that mobility restrictions were particularly effective in the early stages of an outbreak. Ref [16] revealed the relationship between lockdowns and infection cases in Italy and Spain. In the northern parts of Italy, mobility was restricted from March 8, 2020. On March 10, this restriction was imposed throughout the country. Italy implemented two stages of lockdowns. The slope of the increasing rate in daily diagnosed cases, daily deaths, and ICU daily admissions were more stable during the first lockdown, but infection trends continued to rise. After that, a more restrictive lockdown was implemented on March 21. All businesses were closed, except for the essential industries, and the increasing trends of COVID-19 infections changed. Spain imposed similar lockdowns and yielded similar results.

In contrast, several studies show that the first Japanese non-compulsory lockdowns implemented in the spring of 2020 were effective [17–20]. Ref [17] identified the effects of the first Japanese lockdown into the intervention effect and the information effect considering that only the information effect appears in the neighboring prefectures of the prefectures which implemented the voluntary lockdowns. Ref [18] investigated the effects of infection risks and lockdowns on people and their choice to stay at home. Ref [19] quantified the mobility reductions after the first lockdown and showed that human mobility decreased by approximately 50%, and this reduction contributed to a 70% decrease in social contact. They used cellphone data; specifically, Ref [17, 18] used identical data sources. Ref [17] also showed that the first Japanese lockdown had a huge information effect in allowing people to know the infection risk and refrain from going out.

However, most previous studies did not use spatial characteristics of daily trips, such as distance and population density of destinations, in considering the effect of voluntary lockdowns. Although some studies used cellphone data, they only investigated macro-level data such as intercity trips or aggregated trips at the city level. Hence, there are only a few implications about detailed spatial variations in the effects of voluntary lockdowns and infection risks, although they are useful for understanding how and why voluntary lockdowns matter in terms of daily trips. Furthermore, changes in people's attitudes over one year during the pandemic have not been revealed so far.

This study aimed to reveal how human mobility, especially commuting, was affected by the infection risks of COVID-19 and the lockdown policy in the Tokyo metropolitan area

(hereafter, Tokyo MA) in Japan. The novelty of this study is twofold. First, we focus on several characteristics of trips related to infection risks, such as population density and distance, and the variation in the effects of voluntary lockdowns as well as infection risks on daily, intra-city trips by these factors. Our data, collected and estimated from cellphones, are suitable for investigating intra-city trips because they capture how many people who have residency in each city temporarily stay within a 500m × 500m cell each hour. Using the data, we show the effects of infection risks, captured by the number of new infection cases, and that the implementation of lockdowns is more significant if the population density of the destination is large and the distance of the trip is long. This result implies that implementation of lockdowns made people more sensitive to the infection risks; hence, we could show new evidence for the information effect of the policy. Second, our data period was from January 2020 to March 2021. This is a little longer than previous studies and is long enough to capture the change in people's attitudes toward infection risks and the lockdowns over more than a year. We show that the effects of both infection cases and lockdowns became smaller in the latter half of our data period.

## Background

### COVID-19 in Tokyo

The outbreak of COVID-19 was first noticed in Wuhan, Hubei Province, China, and rapidly prevailed worldwide. In Japan, the first infection was confirmed on January 15, 2020. After the first case, the infection spread gradually.

Fig 1 shows the cases of infection in Tokyo and Japan. Both graphs have several waves and exhibit similar trends. In Fig 1(A), we focus on the Tokyo MA. Fig 1(B) shows the number of cases in different prefectures in Tokyo MA. The city has a similar trend to infection cases. In Tokyo, 2,520 people were infected with COVID-19 on January 7, 2021. Further, infection cases increased again around the Olympic Games in Tokyo.

### Lockdowns in Japan and other countries

Many countries have implemented lockdowns in various ways. In this subsection, we show the features of Japanese lockdowns compared to those of other countries. Most countries such as China and Spain imposed compulsory lockdowns and implemented penalties for non-conformists. China imposed strict compulsory lockdowns. In Wuhan, all public transportation via airports, stations, and some roads, was suspended on January 23, 2020. Moreover, the government restricted the population movement. On March 10, 2020, the Chinese government controlled individual activity more strictly using smartphone applications and QR codes, and Chinese people had to register their mobility. However, some countries such as Japan, Sweden, and the United States implemented voluntary lockdowns merely by asking people to refrain from going out.

The United States implemented different NPIs across states, and non-compulsory policies were implemented in some cities such as New York, in which the government announced a state of emergency on March 9, 2020. The government did not restrict private activities such as shopping and going out, but ordered residents not to commute to their office, except for essential workers in health care, social assistance, and the public administration industry. Any company violating this law was fined. In contrast, Japanese and Swedish NPIs asked people to refrain from going out, but were not directly enforceable on individuals. Residents' mobility was not restricted, and the government did not impose penalties. We call such a policy "quasi lockdown" in this paper.

Japan experienced two quasi-lockdowns during our data period, from January 2020 to March 2021. As of July 31, we know that two more lockdowns have been implemented. The

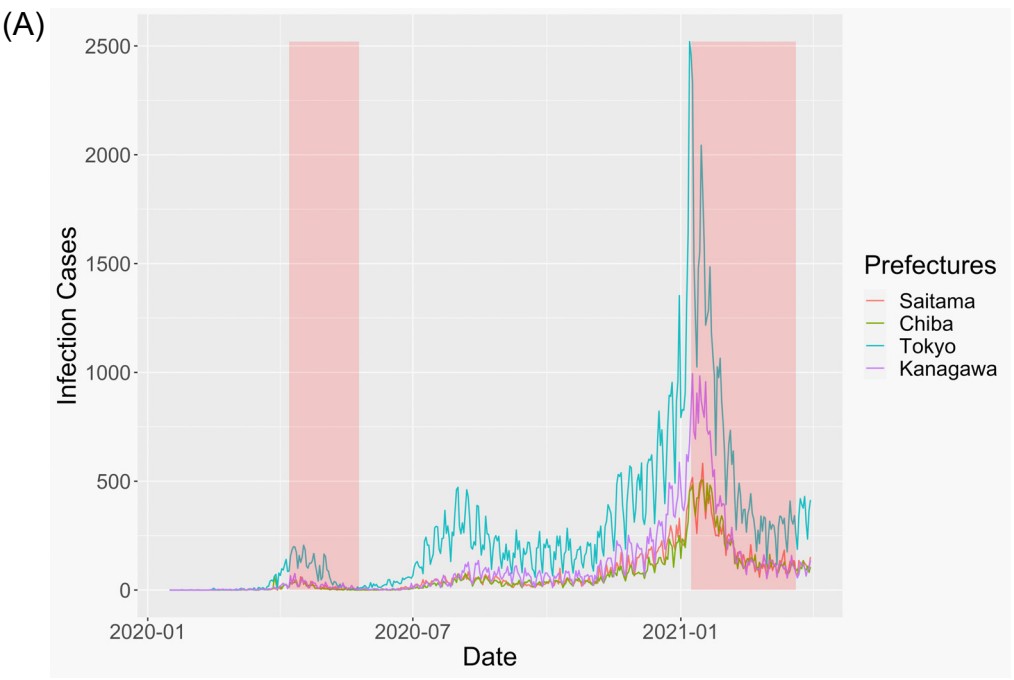

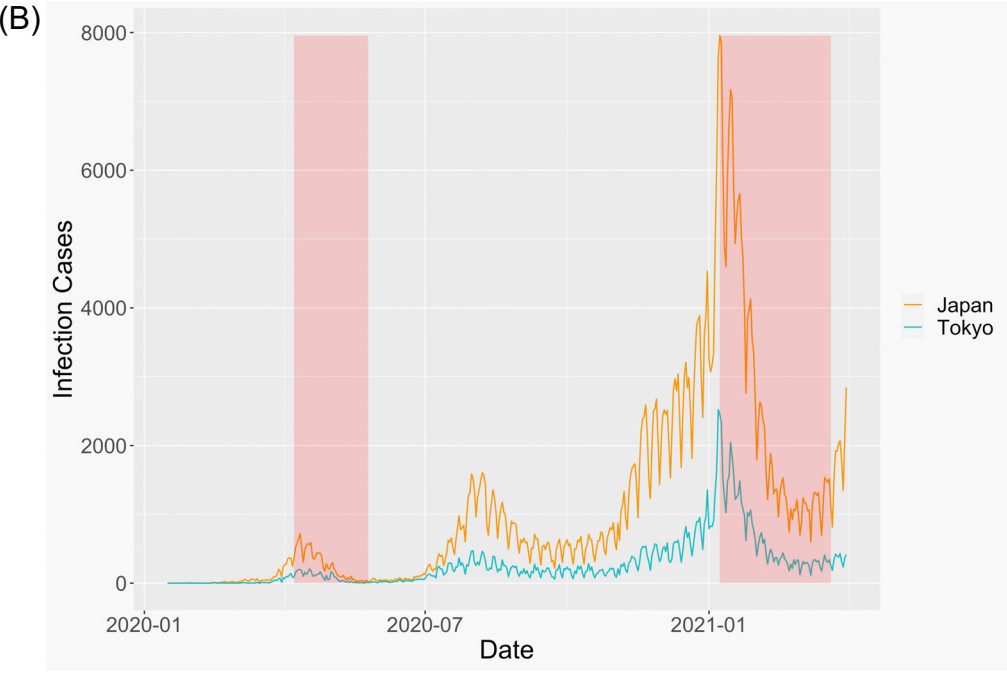

**Fig 1. Infection cases in Tokyo MA (A) and entire Japan (B).**

third was from April 25, 2021, but the affected areas were limited. The end of the periods differ across prefectures. The fourth lockdown started from August 1, 2021, as infection cases increased during the Tokyo Olympics. The first quasi-lockdown was from April 7 to May 25, 2020. From April 7, eight prefectures, including Tokyo, Kanagawa, Chiba, and Saitama prefectures, imposed a quasi-lockdown. From April 16, this lockdown spread nationwide. This

national lockdown ended on May 14, except in Tokyo, Kanagawa, Chiba, Saitama, and Hokkaido prefectures. The lockdown in Tokyo, Kanagawa, Chiba, and Saitama ended on May 25. The second quasi-lockdown continued from January 8, 2021, in Tokyo, Kanagawa, Chiba, and Saitama prefectures to March 21, 2021. During the second quasi-lockdown, other prefectures were added and eliminated.

There are four key features of Japanese lockdowns. First, the Japanese government set the goal of reducing onsite workers by 70%, and many workers started to telecommute. Second, many restaurants with entertainment such as night clubs, hosts, and hostess bars were restricted from opening. If restaurants cooperated with the regulations, they received compensation payments from the government. In the second quasi-lockdown, the government gave more detailed designations regarding restaurants' opening time. Restaurants were requested to close by 8 p.m. and serving alcoholic drinks was limited from 11 a.m. to 7 p.m. In contrast, some libraries, museums, and parks were excluded from this policy. Third, the headcount for large events was at most 5,000 people or 50% of the total capacity. Fourth, the Japanese government set 0.5 infection cases per 100,000 people as a decision criterion for cancelling the quasi-lockdown.

These facts show that no restrictions were imposed directly on the individuals. Although a few restrictions were imposed on some public facilities and the opening of restaurants at night, their effects on daytime mobility, such as commuting, are considered quite indirect and limited.

In this study, we used four prefectures' data in Tokyo MA: Tokyo, Kanagawa, Chiba, and Saitama prefectures, because the infection trend is similar for each prefecture and the period of quasi-lockdown was the same (Fig 1(B)).

## Data

### Data

Three types of data were used in the present analysis. The first is mobility data and mobile spatial statistics data collected from cellular phones. The second is infection data which record the prefecture-level infection cases. The third is geographical data, used to calculate the origin-destination (OD) matrix.

**Mobile spatial statistics data.** We use mobile spatial statistics data provided by DOCOMO Insight Marketing, INC. The data are the estimated hourly population in grid cells of approximately 0.25 km$^2$ based on the location information. The database was collected from the cellular phone users of NTT DOCOMO, INC., one of the largest cellular phone operators in Japan. Mobility data are an expanded estimate of the number of populations from cellular phone users. This database eliminates some grid cells with small population to protect users' privacy. Our data are not individual but aggregate data, and they certainly keep confidentiality so that individuals cannot be identified. And the authors have no access to the raw disaggregated data. We are not involved in the creation, aggregation, or estimation of the data before its distribution. However, as DOCOMO makes this data widely available to users for business and research purposes, it presents the statement in the following Japanese page that DOCOMO complies with the privacy policy regarding the handling of personal data [21].

Mobility data also include information about users' residential cities; hence, the number of people who live in city $i$ and stay at cell $j$ at every hour is available (Fig 2). This study uses the population as the number of trips from city $i$ to the grid cell $j$. However, if the grid cell $j$ is included in city $i$, the remaining population may include people who stay home, so this study excludes such samples from our analysis.

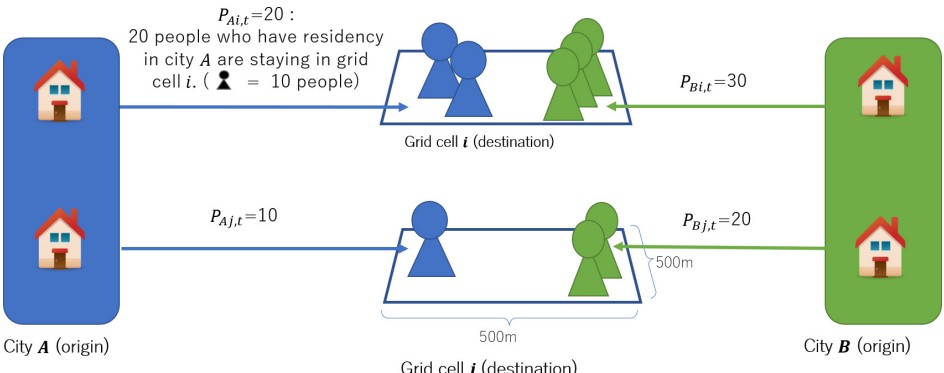

**Fig 2. Outline of mobile spatial statistics data.**

**The number of COVID-19 cases.** We use data from the daily reported new cases of COVID-19 aggregated by NHK [22], which is an acronym for Nippon Hoso Kyokai (Japan Broadcasting Corporation) and the public broadcaster in Japan. This dataset counts new infection cases by prefecture, which are frequently broadcasted as indicators of spreading COVID-19. We use them for independent variables to explain mobility data.

**Geographical data.** We calculated the distance of trips between the center of the grid cells and the residence of samples. We used the location of the city hall of the residential cities instead of the exact residential location. We also employed the Euclidean distance as a proxy of trip distance. Location information of city halls is obtained from the Digital National Land Information [23]. All the location data are included in QGIS [24], and the distances are calculated using R. We used the distGeo package in R that calculates geodesic distance on an ellipsoid.

**The data period and targeted area.** The data period used in our analysis was from January 2020 to March 2021, which includes two lockdowns. Although the original dataset covers over entire Japan, this study uses limited samples in Tokyo MA.

First, our analysis mainly used the population location at 10:00 a.m. Further, to control for the effect of days, we used the data from every Thursday, except for national holidays. Although the objectives of trips are not available in our data, we are especially interested in commuting trips, and measuring the temporary population at 10:00 a.m. is appropriate because most people finish commuting by then. However, we also examined the mobility on holiday afternoons, which is considered mixture of variety of trips.

Second, we only used Tokyo MA because this area experienced two significant lockdowns in our data period. In this study, Tokyo MA is defined to comprise four prefectures: Tokyo, Kanagawa, Chiba, and Saitama. It is different from the definition of the Ministry of Land, Infrastructure, Transport and Tourism (MLIT) based on commuting, which includes parts of other prefectures, while it is not part of those four prefectures. Moreover, since Tokyo MA is the largest metropolitan area in Japan with more than 36 million people, there are a sufficient number of destination grid cells with high population density. Therefore, our dataset includes 60 time points, 28,815 grid cells for destinations of trip, and 244 cities for origins.

## Summary of the mobility data

**Patterns of morning trips.** Our analysis mostly uses the location data of 10:00 a.m. based on the notion that it is the best to capture commuting. However, to check the validity of the idea, we also show the locations at 9:00 a.m. and 11:00 a.m. Fig 3 shows the strong correlation

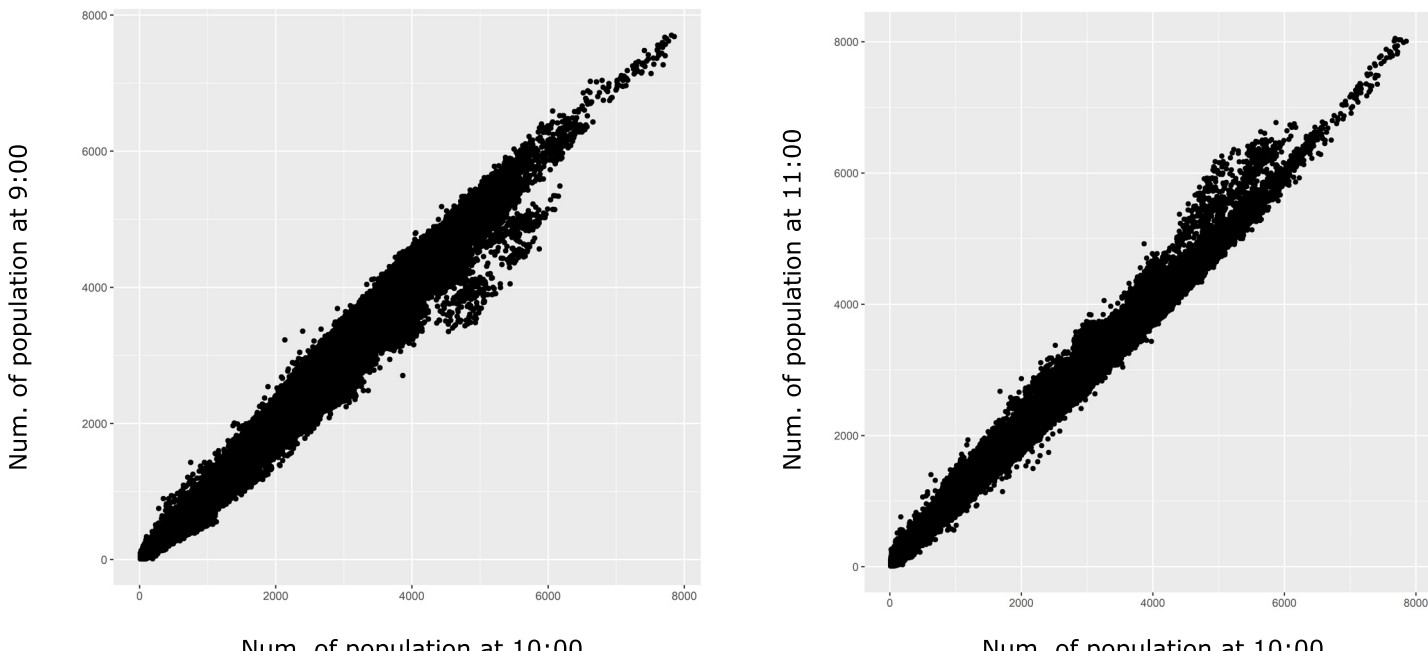

**Fig 3. Comparison of temporary population at 10:00 a.m. to 9:00 and 11:00 a.m.** The data includes the average population on Thursdays in November 2019 in Tokyo MA.

of the number of trips at 9:00 a.m. and 11:00 a.m. with that at 10:00 a.m., which means that the location is almost stable from 9:00 to 11:00 a.m. We can anticipate from the stability that commuting almost finishes by 10:00 a.m., and trips for other objectives are still few in the morning. Therefore, it is supposed that the population location at 10:00 a.m. is mostly the workplace, including students' schools, hence the data at 10:00 a.m. is suitable to capture commuting trips.

**The effect of quasi lockdowns.** We intend to examine the effect of lockdowns. Hence, we will provide an overview of the data to show how human mobility changes during lockdowns.

First, we show how the significance of lockdown effects differs by population density. Fig 4 shows the complementary cumulative distribution of the population in each cell. The blue line indicates the population in each cell at 10:00 a.m. before COVID-19 (November 7, 2019). The red and yellow lines indicate the population during the first (April 9, 2020) and second (March 11, 2021) quasi-lockdowns, respectively. All three days are Thursdays, except for holidays. Dates of both red and yellow lines were chosen for the first Thursday during the lockdown. Fig 4 indicates that change in mobility seems to occur in grid cells with high population density, which seems to be especially significant for a population larger than 4000. Although the share of grid cells with more than 4,000 people in all the populated grid cells is less than 1%, they share 10.4% of the total population in Japan; hence the lockdowns influenced a large number of people.

Further, the black points in Fig 5 show the locations of highly populated places. Most of these points are concentrated in large metropolitan areas such as Osaka and Chubu, and especially Tokyo. This research, therefore, focuses on the Tokyo MA because it experienced two lockdowns and had a sufficient number of sample grid cells affected by them.

Fig 6 visualizes the change in staying population by the lockdowns in the Tokyo MA by comparing the average population of each cell between lockdown periods and before COVID-19 (S1 Video). We took the average number of people every Thursday from November 7 to

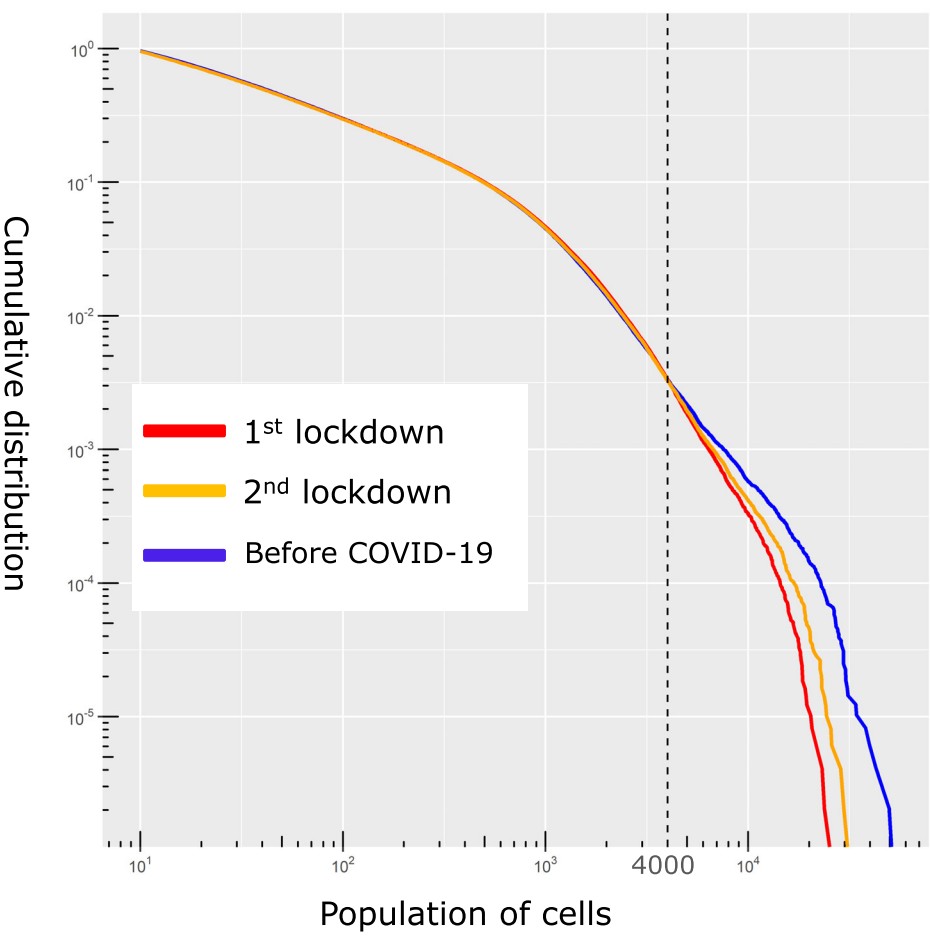

**Fig 4. Complementary cumulative distribution function of population in approximately 500m × 500m grid cells.**

December 19, 2019. We omitted December 26 from the average population. During both lockdowns, we see a significant decrease in mobility in the central area of Tokyo, which is 30% larger than the first one. The effect is smaller in subcenters and suburban areas, and the temporary population in the surrounding areas increases as a result of the stay home movement. Although they might imply that long-distance trips to high-density places decrease more significantly, we need further statistical investigation to show that. Further, they also show that the effect of the first lockdown is more significant than that of the second one.

## Methods

We used panel data to conduct a fixed-effect estimation. The equation to estimate is

$$\ln P_{ij,t} = \alpha_{ij} + \alpha_t + \beta_1 \mathbf{X}_{ij,t}, \tag{1}$$

where $P_{ij,t}$ is the number of people moving from residential city $j$ to the destination grid cell, $i$, on day $t$. Since our primary interest is commuting behavior, the data at 10:00 a.m. on every Thursday, except for national holidays, is used. $\alpha_{ij}$ is the fixed effect determined by the OD pair, $i$, and $j$. $\alpha_t$ is the fixed effect of each date $t$.

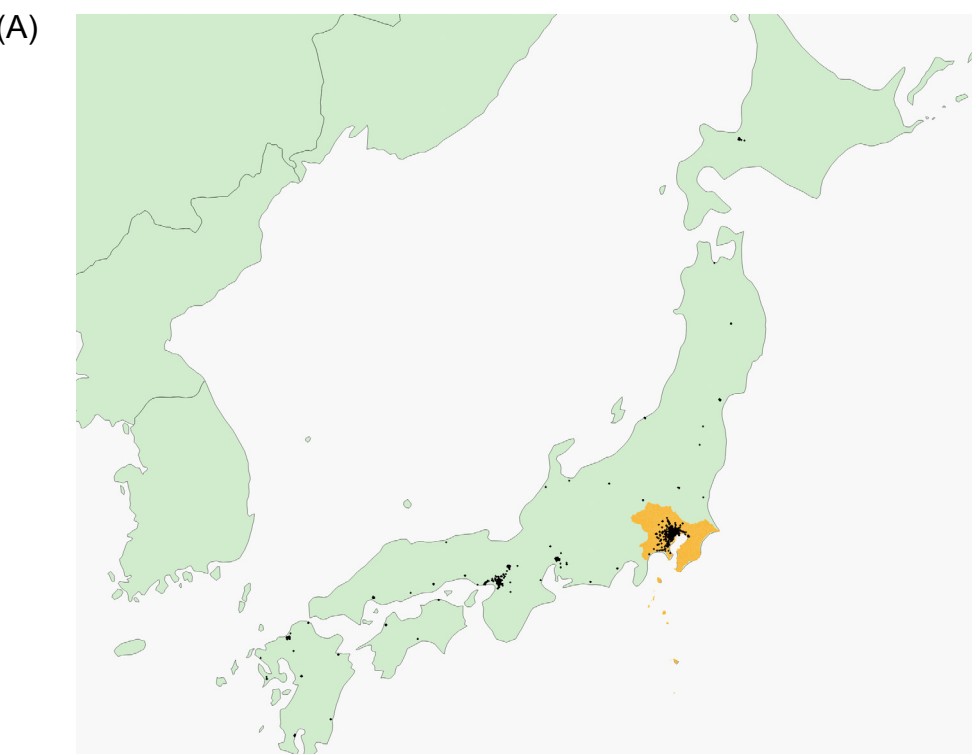

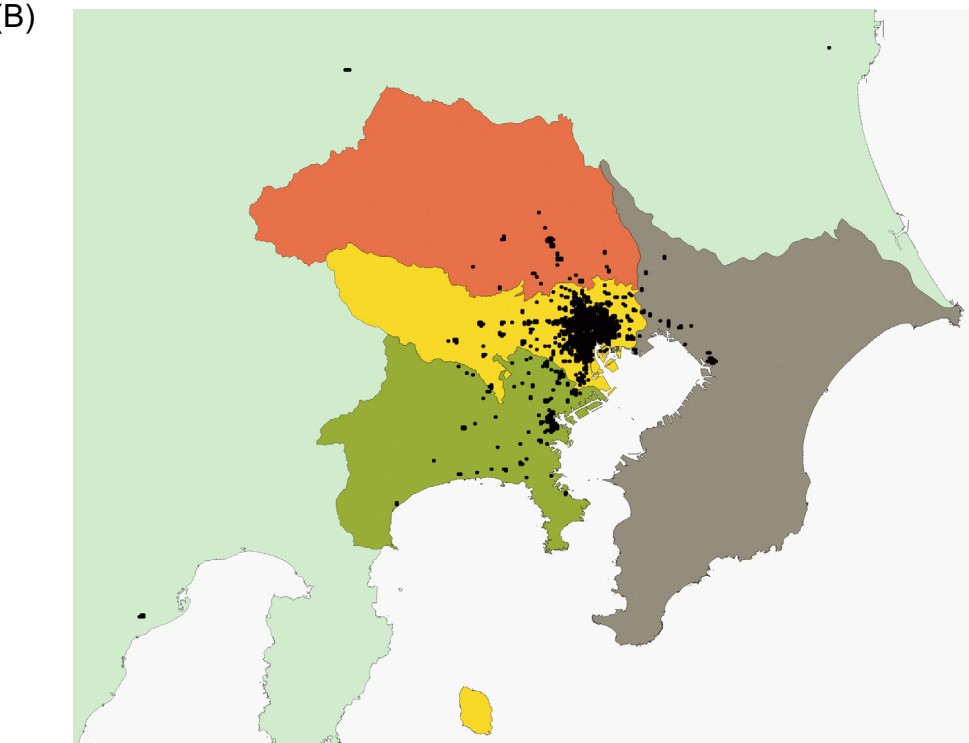

**Fig 5. Location of high-density places.** (A) Black dots in the map represent grid cells in which the average population at 10:00 a.m. on Thursdays in November 2019 exceeded 4,000. The yellow area represents Tokyo MA. (B) We analyze four prefectures: The deep green is Kanagawa prefecture, the yellow is Tokyo prefecture, The orange is Saitama prefecture, and the gray is Chiba prefecture. We obtained Figs 5A and 5B from Natural Earth and National Land Information Division, National Spatial Planning and Regional Policy Bureau, Ministry of Land, Infrastructure, Transport and Tourism of Japan.

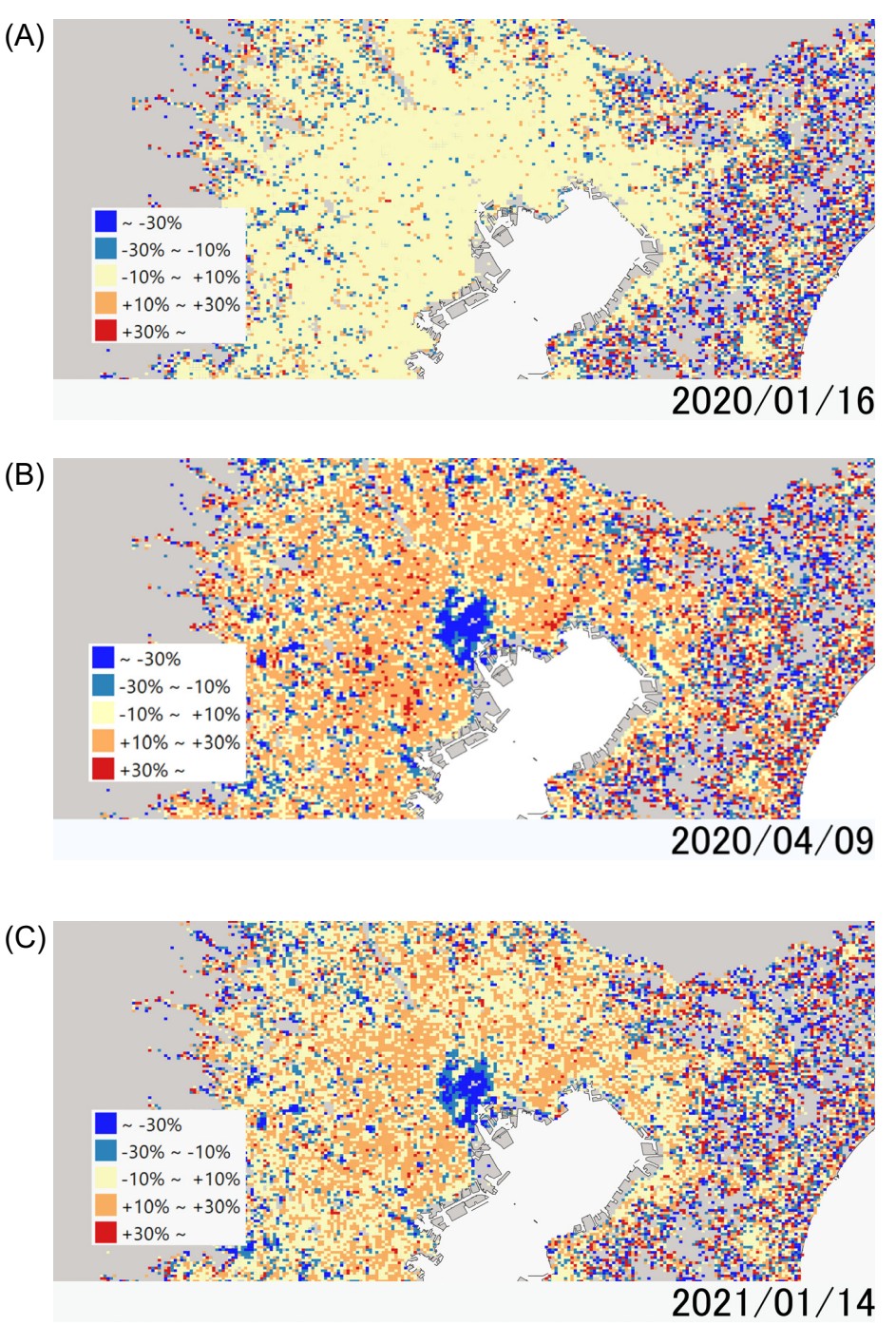

**Fig 6. Population density in Tokyo MA in comparison to before COVID-19: (A)Before COVID-19; (B)First lockdown; (C) Second lockdown.** We obtained Figs 6A, 6B, and 6C from Natural Earth and National Land Information Division, National Spatial Planning and Regional Policy Bureau, Ministry of Land, Infrastructure, Transport and Tourism of Japan.

$\mathbf{X}_{ij,t}$ is the vector of the independent variables as follows: First, we used the relative(i.e. per-population) number of infection cases from one week ago (*IFC*) in the destination prefecture to show infection risk by trip. The motivation of considering the relative infection cases is to control the difference in population sizes among prefectures for evaluating substantial

infection risk. For that purpose, we define *IFC* as number of infection cases per relative population size in 2020, which describes the ratio of residential population size of each prefecture to Tokyo; hence, that of Tokyo is exactly one. According to 2020 population census, we use 14, 064, 696 in Tokyo, 9, 240, 411 in Kanagawa, 6, 287, 034 in Chiba, and 7, 346, 836 in Saitama as the number of population.

We consider time lags of reaction for COVID-19 spreading in these indices. For example, some onsite workers change their commuting patterns in the next week after watching news about COVID-19 on weekends. To capture this time lag, we use data from Monday to Sunday in weekday analysis, and from Sunday to Saturday in weekend analysis. Following the same idea, we also used the number of infection cases two weeks prior to that of one week ago (*LONG*) to describe how the recent (short-term) outbreak is prolonged. If this index is large, people will have enough time to change their behavior accordingly.

Second, $\mathbf{X}_{ij,t}$ also includes the lockdown dummy variables that take 1 during each of the first and second lockdowns in the Tokyo MA; they are implemented from April 7 to May 25, 2020, and from January 8 to March 25, 2021, respectively.

Third, $\mathbf{X}_{ij,t}$ also includes the cross terms of the above variables, the number of infections and lockdown dummies, the population density of destination cells, and trip distance. Detailed calculation is given in Section titled Total effects and their sensitivity. These cross terms capture different reactions to the virus, prevailing by the characteristics of trips. We use the population in the sample grid cell *i* before COVID-19 as the population density. Single terms of *Dens* and *Dist* are excluded because they are included in the fixed effects of OD. In the same reason, *LD*1 and *LD*2 are excluded when the date fixed effects are estimated.

Finally, note that *IFC*, *Dens*, *Dist*, and *LONG* actually takes the forms of $X_{ij,t} = \ln(1 + x_{ij,t}/\omega^x)$ in Eq (1) to employ the log-linear relationship while treating the case that their value may take zero. In what follows, variables written with lowercase letters, such as $ifc_{ij,t}$, $dens_{ij}$ and $dens_{ij}$, denote the values before taking the log; hence, $IFC = \ln(1 + ifc_{ij,t}/\omega^x)$ holds, for example. Here $\omega^x$ is unit of each variable, which is chosen to maximize the value of within $R^2$ in the Model 2 of the baseline estimation presented in Table 1. Because of the computational burden of our estimation using the large size dataset, we could examined only 27 combination patterns for $\omega^x$: {0.5, 1, 10} for *IFC*, and {1, 100, 1000} for *Dens* and *Dist*. Since *LONG* is originally the ratio between two weeks, the unit can be normalized to one. $(\omega^{IFC}, \omega^{Dens}, \omega^{Dist})$ = (0.5, 1000, 1) is finally chosen and the within $R^2$ is 0.13074, as noted in Table 1. On the other hand, the worst case is (10, 1, 1000) whose within $R^2$ is 0.12267; hence, the effect of the unit on the model's performance is not very large. Furthermore, for checking the robustness of our result regarding the choice of functional form, we also examine the functional form $IFC_{ij,t} = \ln(ifc_{ij,t} + \sqrt{1 + ifc_{ij,t}^2})$, which was employed in Ref [17]. Although the signs of the estimates are almost the same as those in Table 1, within $R^2$ is 0.1273 and lower than our optimal model.

## Results

### Baseline estimation

Our panel estimations of Eq (1) are reported in Table 1. Model 1 shows the result of the simple panel estimation without cross terms. Both *IFC* and *LONG* were negatively correlated with mobility, showing that commuting decreased as infection cases increased. From the result, a 1% increase in *ifc* decreases mobility by 1.01% given the value of *ifc* is sufficiently large. For simplicity, we calculate the elasticity by using $\ln(ifc/\omega^{IFC})$ instead of $\ln(1 + ifc/\omega^{IFC})$, which

**Table 1. Baseline results.**

| | Model 1 | Model 2 | Model 3 | Model 4 |
|---|---|---|---|---|
| IFC | −0.0101*** | 0.0308*** | −0.0225*** | 0.0173*** |
| | (0.0001) | (0.0013) | (0.0003) | (0.0014) |
| IFC×Dens | | −0.0146*** | | −0.0147*** |
| | | (0.0001) | | (0.0001) |
| IFC×Dist | | −0.0022*** | | −0.0025*** |
| | | (0.0001) | | (0.0001) |
| LD1 | −0.2192*** | 0.0484*** | | |
| | (0.0007) | (0.0096) | | |
| LD1×Dens | | −0.1326*** | | |
| | | (0.0007) | | |
| LD1×Dist | | −0.0090*** | | |
| | | (0.0011) | | |
| LD2 | −0.0028*** | 0.0780*** | | |
| | (0.0003) | (0.0043) | | |
| LD2×Dens | | −0.0045*** | | |
| | | (0.0004) | | |
| LD2×Dist | | −0.0084*** | | |
| | | (0.0005) | | |
| LONG | −0.0143*** | 0.0051*** | −0.0163*** | 0.0238*** |
| | (0.0001) | (0.0014) | (0.0002) | (0.0014) |
| LONG×Dens | | −0.0167*** | | −0.0217*** |
| | | (0.0001) | | (0.0001) |
| LONG×Dist | | 0.0006*** | | −0.0011*** |
| | | (0.0002) | | (0.0002) |
| FE : OD | YES | YES | YES | YES |
| FE : date | NO | NO | YES | YES |
| $R^2$ | 0.9349 | 0.9381 | 0.9377 | 0.9408 |
| Adj.$R^2$ | 0.9321 | 0.9354 | 0.9350 | 0.9382 |
| within.r.squared | 0.0864 | 0.1307 | 0.0011 | 0.0507 |
| Observations | 10, 060, 792 | 10, 060, 792 | 10, 060, 792 | 10, 060, 792 |

Notes: Figures in parentheses are cluster-robust standard errors by OD.

***, **, and * denote statistical significance at the 0.1%, 1%, and 5% level, respectively.

means that we assume infinitely large *ifc*. However, since $\omega^{IFC}$ is set 0.5, the error from using the approximation is less than 0.1% of the elasticity value when we assume *ifc* = 100. Lockdown dummies *LD*1 and *LD*2 also have statistically significant negative effects, but the magnitude of the effect of *LD*2 is smaller. Note that this analysis only considers short-run changes, but there will be some long-run effects if such increases in infection cases last for a long time because people may change their lifestyles fundamentally. Additionally, the implementation of the first and second lockdown decreased mobility by 21.92% and 0.28%, respectively. Note that LDs are dummies and hence take the values 1 or 0 while *ifc* is a continuous variable whose natural logarithm is used for the independent variable. Therefore, when *LD*1 changes from 0 to 1, the mobility changes by $\beta_{LD1} \times 100\%$, where $\beta_{LD1}$ is the coefficient of *LD*1. This study also shows a large influence of the Japanese voluntary lockdown, as in former studies, as well as a statistically significant but weak effect of the second lockdown.

Model 2 includes cross terms for the two types of infection data and lockdown dummy variables for commuting distance and population density. All cross terms related to infection cases are negatively correlated with mobility. Understanding these results may be complicated. We explain the cross terms using *IFC×Dens* as an example. When we focus on one day, which means that we set *IFC* fixed, then the negative coefficient shows that the higher population density grid cells have, the fewer people commute to such grid cells as the destination. Therefore, the result implies that people avoid long commutes and crowded places when infection cases increase.

This behavior is rational because when the number of infected people increases, the risk of infection in crowded places where there is a lot of contact between people increases significantly. The risk of infection is generally determined by the number of contacts with the infected people, which is positively correlated with the number of contacts with others and the share of the infected people in the population. This is why risk increases significantly in congested places when the number of infection cases increases. This is also the case for long-distance trips where people are exposed to the risk in congested trains for a long time. However, this result is also explained by the large incentive to introduce telecommuting for people who commute long distance. When the pandemic made teleworking more socially acceptable, such people were more likely to choose telework to save their large commuting costs.

The cross terms related to the lockdowns also show negative coefficients for the first lockdown, while the effect of density is ambiguous for the second lockdown. That is, people avoid long-distance trips and crowded places when lockdowns are imposed; a similar reaction is seen when there is an increase in infection risks, described by the number of infection cases. This similarity can be explained by the information effect of lockdowns presented by [17]. The governments of Japan and Tokyo decided to impose this lockdown considering the increasing infections, and it was widely known through the media; hence, people accepted the lockdowns as warnings against the risk of infection. Although we do not identify the information effect from other effects such as the intervention, unlike [17], our results provide additional evidence for the existence of the information effect of the voluntary lockdown.

Furthermore, one may consider that the coefficient of the single term of *IFC* is positive, and mobility might increase when the number of infection cases increases. However, infection also decreases the mobility via the cross terms, and the total marginal effect is negative, as examined in the later part.

Finally, we introduce time-fixed effects in Models 3 and 4, where the lockdown dummies must be omitted because they are included in the time-fixed effects. Despite the control of an additional fixed effect, most estimates do not change drastically; hence, most of our results are robust.

## Total effects and their sensitivity

Our key variables, number of infections and lockdowns, affect mobility via multiple cross terms in Model 2. Since the total marginal effects differ depending on the distance of trips and density of destination, we need to determine how much the effect varies by them. In addition, every single coefficient does not simply tell us the total influence quantitatively when they change, and we need to evaluate these variables by their total effects. Therefore, we calculated the total effects and their sensitivities to density and distance.

## Total marginal effects of infection cases (IFC)

**Fig 7. Total marginal effects of infection cases (*IFC*).** Population density and trip distance are fixed to 2000 and 10km when the other variable is changed.

To quantify the total marginal effect, we define the elasticity of the mobility change as follows:

$$e = \frac{\frac{d\,P}{P}}{\frac{difc}{ifc}} \simeq \frac{d\ln P}{dIFC}. \tag{2}$$

This elasticity describes what percent of mobility decreases when infection cases *ifc* decrease by one percent. Note that, again, we assume sufficiently large *ifc*. From Eq (1), we can calculate *e* as follows:

$$e = \beta_0 + \beta_1 Dist + \beta_2 Dens, \tag{3}$$

where $\beta_0$, $\beta_1$, and $\beta_2$ denote the coefficients of *IFC*, *IFC* × *Dens*, and *IFC* × *Dist* obtained from Model 2, respectively; hence, *e* is called the total effect of *IFC*. This total effect depends on the values of the distance and population density. Therefore, we use specific values to check whether mobility increases or decreases in specific areas. Fig 7 shows the total marginal effects of *IFC*, where dots represent the estimated values and bars represent 95% confidence intervals. They show that the total effect of *IFC* is negative in most cases in Fig 7, except for a case with small population density. Moreover, the total effects are highly sensitive to the population density and trip distance. Suppose the distance of the destination is fixed at 10 km and the population density is changed from 2,000 to 8,000; the total effect of infection cases ranges from -0.02 to -0.005, which is the difference of about four times. That is, the effect of infection cases

largely varies in the possible range of density and distance, and hence, their effects are significant.

We should also note that the predicted total effect of *IFC* may be positive significant for about 17% of our OD samples whose distance and population density are small. Although this might looks somewhat counter-intuitive, it may be related to the change in mobility pattern during COVID-19 pandemic reported in several previous studies. People are more likely to use less congestible transport modes (e.g., cars, bicycle, and walk rather than public transport) as reported by Ref [25] and go to near and low density places [10, 15, 26], and mobility to parks significantly increases after lockdowns ends even when the infection cases remained high as in Ref [15]. It is also reported by Ref [20] that prefecture-level mobility unchanged or slightly increased in several prefectures in Japan whose population density and number of infection cases are relatively few.

We also calculated the sensitivity of the total effects of the two lockdowns. Fig 8 shows that the total effect of the first lockdown also largely differs depending on the density and distance, as in *IFC*, although the lockdowns' effects are slightly less sensitive than those of *IFC*. Owing to the difference in sensitivity of the cross terms, the total effects of the two lockdowns are strongly negative, and no significant positive effect is predicted for any OD samples unlike *IFC*. The lockdowns strictly mitigate the trips even if they are a short distance or their destination has low population density.

These results may be reasonable considering that, following [17], lockdowns have two different effects: one is the information effect that lets people know the risks, and the other is the intervention effect to control people's mobility with legal or psychological coercion. From our result, the negative coefficients of cross terms support the former effect, while the strict negative total effect to all OD samples support the latter effect. The significance of both effects was also shown by [17]. Therefore, if governments are aware of the role of that effect in sending an effective message to people, they can use this political tool more effectively.

However, the results for the second lockdown were weak and ambiguous, implying that it was less effective. Since Japan experienced at least two more lockdowns by July 31, 2021, after the data period, some additional examinations will be necessary to identify how that effect is still effective after the first lockdown.

## Trend of trip patterns

During just one year after the pandemic broke out, there were various changes in our lifestyles and surroundings, which may affect the attitudes of people to infection risk and lockdowns. Various events such as dissemination of COVID-19's infection mechanisms, adaptation to new behavioral patterns, vaccine development reports, and emergence of mutant strains are occurring. Although we cannot examine each of them individually, we attempt to describe what happens in people's hidden attitude that is not explained by our basic estimation.

We begin with the date fixed effect because it is useful to reveal the trend of basic trip frequency of people in each time period, which is not fully explained by any independent variables in the model. Fig 9 represents the estimates of the date fixed effect estimated in Models 3 and 4 in Table 1, which shows two notable changes. First, there is a significant decrease around April of 2020, which represents the effects the first lockdown included in the date fixed effects. A very significant but temporal decrease on August 13 merely shows the week of Japanese traditional yearly vacation (called Obon). Again, note that the date fixed effects and lockdown dummies are not separable in our dataset.

Second, relatively high activeness continues during several months in autumn. It implies that people became used to the pandemic and less sensitive through dissemination of the basic

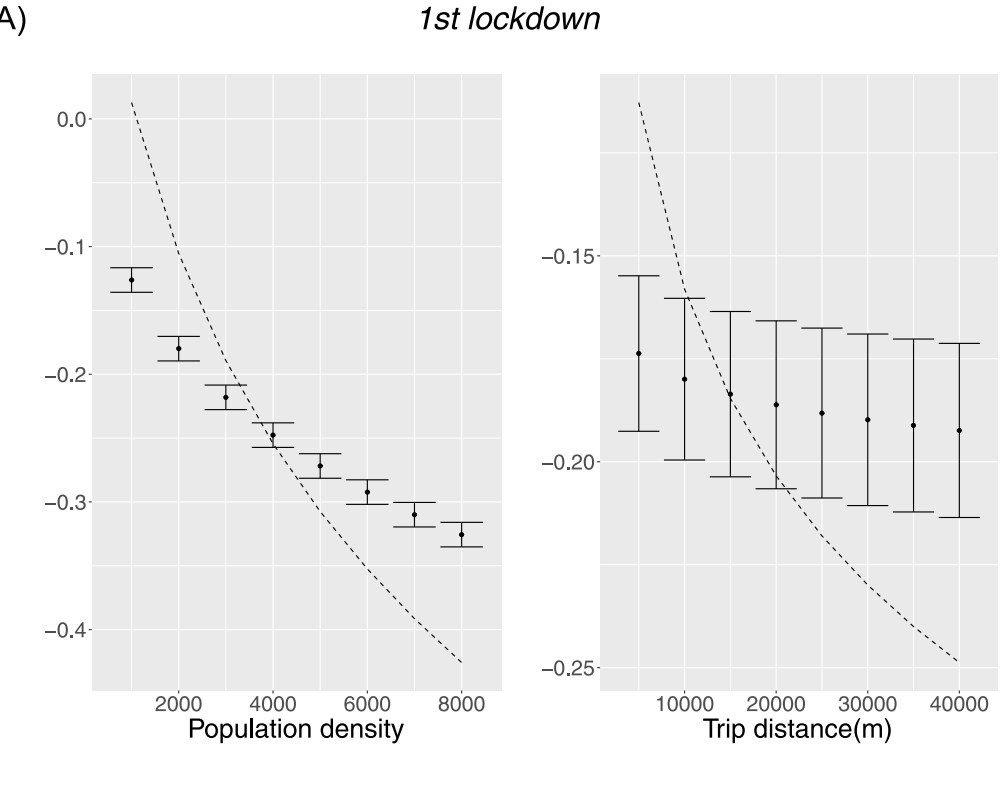

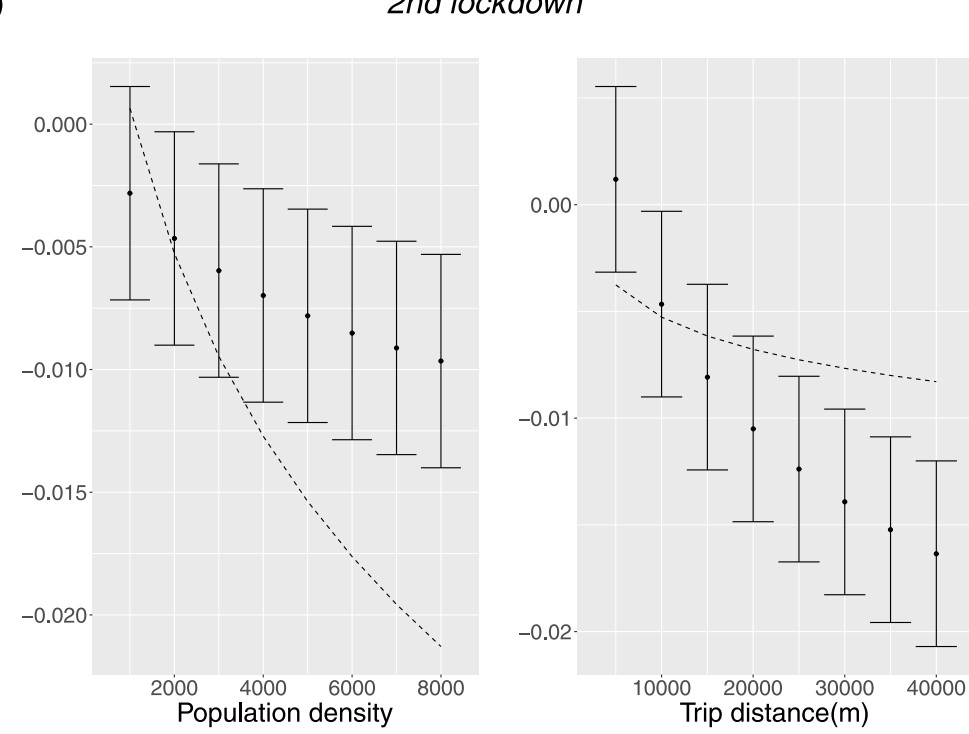

**Fig 8. Total marginal effects of the lockdowns: (A)First lockdown; (B) Second lockdown.** The total marginal effects of each lockdown are described by $\beta_{0m} + \beta_{1m}Dist + \beta_{2m}Den$, where $\beta_{0m}$, $\beta_{1m}$, and $\beta_{2m}$ represent the coefficients of the single and cross terms of the mth lockdown in Model 2 of Table 1. The dashed lines show the total effects of the *IFC* under the same conditions. Those with the first lockdown are multiplied by 20 in the left, Population density, and 30 in the right, Trip distance (m). However, no scale adjustment is performed for the second lockdown.

(A)

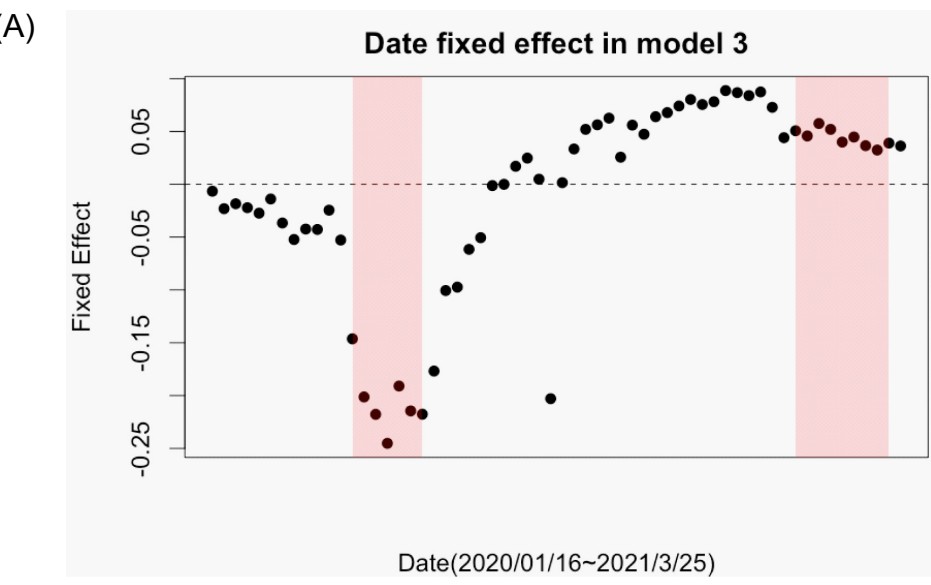

(B)

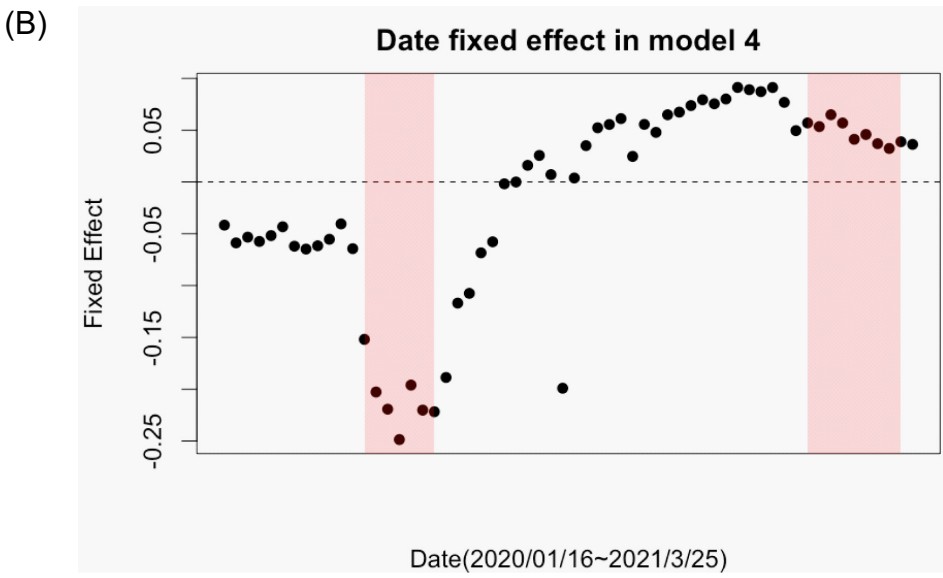

**Fig 9. Date fixed effect: (A) in Model 3 and (B) in Model 4.** In both figures, date fixed effect significantly decreases around April, 2020.

knowledge about COVID-19 risks six months after the break out. However, the fixed effect decreases again as the second lockdown was announced.

Further, we conducted additional estimations that divide the data into five periods, where each length is about 12 weeks, and estimate each data period separately. Although our baseline estimation assumes that the value of each parameter is constant throughout the data periods, such additional estimation allows us to capture how the attitude of people toward the infection risks changed through the change in parameters across periods.

Table 2 shows that the basic properties of estimation results are stable and almost consistent to the baseline long-term estimation. Nevertheless, because of various period-specific situations and specificity of short-run behavioral change, magnitudes of coefficients vary across periods and some of them even show the opposite signs to the basic hypothesis. The positive

**Table 2. Estimation results for detailed periods.**

| | Period 1 | Period 2 | Period 3 | Period 4 | Period 5 |
|---|---|---|---|---|---|
| IFC | 0.02287*** | 0.06246*** | 0.00864 | 0.04767*** | −0.01764*** |
| | (0.00193) | (0.00412) | (0.00459) | (0.00403) | (0.00354) |
| IFC×Dens | −0.00988*** | −0.01691*** | −0.01016*** | −0.00729*** | −0.01244*** |
| | (0.00018) | (0.00038) | (0.00038) | (0.00040) | (0.00034) |
| IFC×Dist | −0.00390*** | −0.00829*** | −0.00150** | −0.00479*** | 0.00252*** |
| | (0.00022) | (0.00048) | (0.00052) | (0.00047) | (0.00041) |
| LD1 | | −0.03875*** | | | |
| | | (0.00771) | | | |
| LD1×Dens | | −0.06564*** | | | |
| | | (0.00072) | | | |
| LD1×Dist | | 0.00438*** | | | |
| | | (0.00089) | | | |
| LD2 | | | | | 0.08443*** |
| | | | | | (0.00516) |
| LD2×Dens | | | | | −0.02023*** |
| | | | | | (0.00051) |
| LD2×Dist | | | | | −0.00625*** |
| | | | | | (0.00060) |
| LONG | −0.01368*** | 0.06594*** | −0.01670** | −0.02217** | −0.01379** |
| | (0.00175) | (0.00290) | (0.00588) | (0.00725) | (0.00525) |
| LONG×Dens | −0.00814*** | −0.03008*** | −0.00429*** | −0.01048*** | −0.00205*** |
| | (0.00017) | (0.00027) | (0.00045) | (0.00072) | (0.00051) |
| LONG×Dist | 0.00090*** | −0.00773*** | 0.00309*** | 0.00289*** | 0.00083 |
| | (0.00020) | (0.00034) | (0.00066) | (0.00084) | (0.00061) |
| FE : OD | YES | YES | YES | YES | YES |
| FE : date | NO | NO | NO | NO | NO |
| $R^2$ | 0.96361 | 0.96031 | 0.96046 | 0.96695 | 0.96534 |
| Adj. $R^2$ | 0.95740 | 0.95415 | 0.95375 | 0.96222 | 0.95942 |
| within.r.squared | 0.05481 | 0.22575 | 0.00409 | 0.00082 | 0.00565 |
| Observations | 2004529 | 1898701 | 1883602 | 2303607 | 1843357 |

Notes: Figures in parentheses are cluster-robust standard errors by OD.

***, **, and * denote statistical significance at the 0.1%, 1%, and 5% level, respectively.

We divide the data into five periods; from January 2020 to March 2020, from April 2020 to June 2020, from July 2020 to September 2020, October 2020 to December 2020, and January 2021 to March 2021. We omit August 13 because it is during Japanese vacation and the mobility is specially low as seen in Fig 9.

sign of $LD1 \times Dist$ in the second period can be explained by the fact that short distant trip recovered more flexibly after lockdowns end [15]. This can also explain the positive sign of $IFC \times Dist$ in the fifth period; because of such flexibility of short-distant trips, they may also decrease rapidly again when the infection cases increased in the winter of 2020–2021.

Since it is difficult to mention each of them in detail, we simply summarize them into the total marginal effects and see their trend presented in Fig 10. First, although five periods are too few to conclude, the trend of the total marginal effects of *IFC* look somewhat similar to that of the date fixed effects. We can also read the same implication behind them; that is, people had been highly cautious about the infection risks in the early stage of the pandemic (i.e. around the first lockdown), and they were getting used to the new situation after the summer

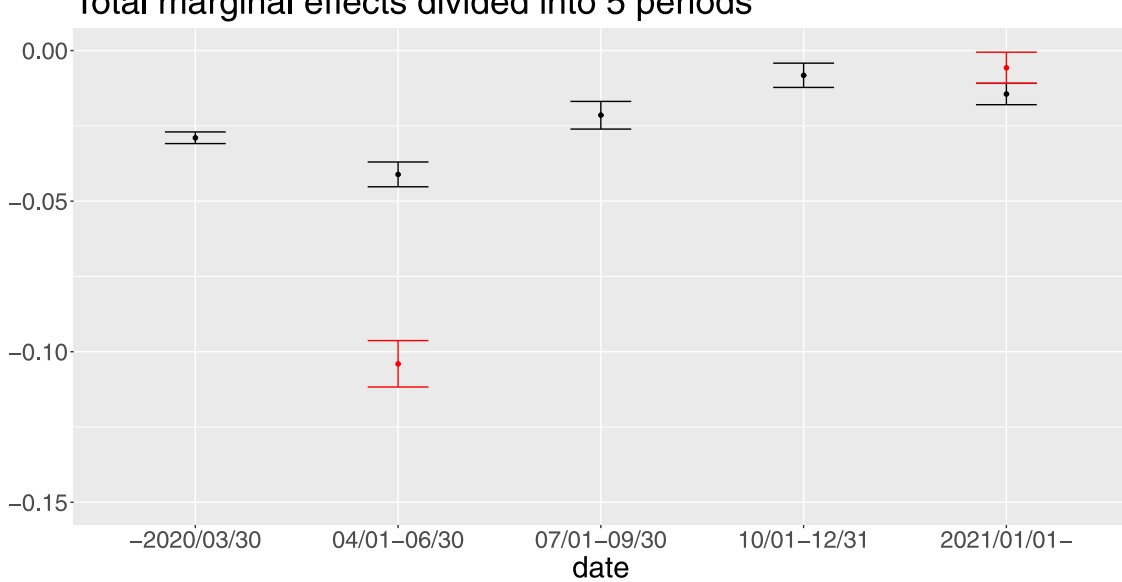

**Fig 10. Total marginal effects divided into five periods.** We also calculate total marginal effect as for *IFC*, *LD*1, and *LD*2 by Eq 3.

of 2020. Second, as reported in the baseline result, the effect of the second lockdown is quite small in comparison to the first one.

## Mobility in weekend afternoon

Although this study has mainly focused on weekday mornings to investigate commuting trips, we also conducted the same analysis using the data from the weekend afternoon (14:00), which is considered to include trips for personal purposes such as leisure, shopping, and commuting. The results are reported in Table 3 and are similar to weekday mornings; hence, most of our implications can be extended to various types of trips. The unit of *IFC*, *Dens*, and *Dist* are also estimated to be 10,1, and 1000, respectively. However, the results about *LONG* somewhat differ; the effect of the downward trend of infection cases is positive in Model 1, and the cross term with Dens is positive in Model 2. Although the negative effect in commuting trip stands for enough adjustment time, this result show that the opposite effect from the stress of long-lasting self-restraint might be rather dominant in the private trips. However, this is still ambiguous because Models 3 and 4 show the opposite result, and we need to implement further estimations to investigate such controversial results.

## Conclusion

We investigated how the effects of infection risks and voluntary lockdowns on mobility differ by distance and density using mobility data with detailed spatial characteristics, and the following three main results were obtained. The first is the mobility reduction caused by infection cases. People decrease their mobility after getting information about infection cases. The reduction of mobility increases with the population density of destination and the distance of the trip. This means people avoid the risk of infection considering the spatial characteristics of trips. Second is the information effect of the non-compulsory lockdowns. People avoided congested places and long-distance trips during lockdowns as in the case of an increase in infection cases, which provides evidence that the information effect of the non-compulsory lockdowns alerts people to the infection risks. Third is the change in reaction of people during

**Table 3. Estimation results for the weekend afternoon.**

| | Model 1 | Model 2 | Model 3 | Model 4 |
|---|---|---|---|---|
| IFC | −0.0096*** | 0.0375*** | −0.0639*** | −0.0105*** |
| | (0.0001) | (0.0023) | (0.0007) | (0.0024) |
| IFC×Dens | | −0.0125*** | | −0.0099*** |
| | | (0.0002) | | (0.0002) |
| IFC×Dist | | −0.0033*** | | −0.0024*** |
| | | (0.0003) | | (0.0003) |
| LD1 | −0.3003*** | 0.7194*** | | |
| | (0.0015) | (0.0165) | | |
| LD1×Dens | | −0.3398*** | | −0.3284*** |
| | | (0.0016) | | (0.0016) |
| LD1 ×Dist | | −0.0676*** | | −0.0700*** |
| | | (0.0019) | | (0.0019) |
| LD2 | −0.0119*** | 0.0625*** | | |
| | (0.0004) | (0.0055) | | |
| LD2×Dens | | −0.0096*** | | −0.0164*** |
| | | (0.0006) | | (0.0006) |
| LD2 × Dist | | −0.0076*** | | −0.0054*** |
| | | (0.0006) | | (0.0006) |
| LONG | 0.0468*** | 0.0968*** | −0.0419*** | −0.0067 |
| | (0.0004) | (0.0052) | (0.0005) | (0.0051) |
| LONG×Dens | | 0.0552*** | | 0.0438*** |
| | | (0.0005) | | (0.0005) |
| LONG×Dist | | −0.0125*** | | −0.0063*** |
| | | (0.0006) | | (0.0006) |
| FE : OD | YES | YES | YES | YES |
| FE : date | NO | NO | YES | YES |
| Observation | 6206073 | 6206073 | 6206073 | 6206073 |
| Num. groups: OD | 372284 | 372284 | 372284 | 372284 |
| Num. groups: date | | | 59 | 59 |
| $R^2$ | 0.9184 | 0.9238 | 0.9231 | 0.9280 |
| Adj. $R^2$ | 0.9132 | 0.9189 | 0.9182 | 0.9234 |
| within $r^2$ | 0.0678 | 0.1293 | 0.0039 | 0.0676 |

***$p < 0.001$; **$p < 0.01$; *$p < 0.05$

Notes: Figures in parentheses are cluster-robust standard errors by OD. ***, **, and * denote statistical significance at the 0.1%, 1%, and 5% level, respectively.

one year. Our results show that people react less sensitively to infection cases in the later periods (September 2020–March 2021) compared to the early periods (January 2020–July 2021), and the second lockdown was less effective than the first.

## Supporting information

**S1 Video. The change in staying population in the Tokyo MA.**
(MP4)

## Acknowledgments

We are grateful to all seminar and conference participants at the XV World Conference of Spatial Econometrics Association (SEA 2021), The 11th Asian Conference In Regional Science,

and the 35th Applied Regional Science Conference for their useful comments and discussions. Keisuke Kawata, Ștefana Cioban, Daisuke Fukuda, Giuseppe Arbia, Dao-Zhi Zeng, Tatsuhito Kono, and Tomokatsu Onaga also gave us valuable comments.

## Author Contributions

**Conceptualization:** Ryo Itoh.

**Data curation:** Kazufumi Tsuboi, Naoya Fujiwara.

**Formal analysis:** Kazufumi Tsuboi.

**Funding acquisition:** Naoya Fujiwara.

**Investigation:** Kazufumi Tsuboi, Naoya Fujiwara.

**Methodology:** Kazufumi Tsuboi, Naoya Fujiwara, Ryo Itoh.

**Project administration:** Ryo Itoh.

**Supervision:** Ryo Itoh.

**Validation:** Kazufumi Tsuboi.

**Visualization:** Kazufumi Tsuboi, Ryo Itoh.

**Writing – original draft:** Kazufumi Tsuboi.

**Writing – review & editing:** Naoya Fujiwara, Ryo Itoh.

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
