## [Decision Letter · Decision Letter 0]

14 Feb 2022

PONE-D-21-40257Influence of trip distance and population density on intra-city mobility patterns in Tokyo during COVID-19 pandemicPLOS ONE

Dear Dr. Tsuboi,

Thank you for submitting your manuscript to PLOS ONE. After careful consideration, we feel that it has merit but does not fully meet PLOS ONE’s publication criteria as it currently stands. Therefore, we invite you to submit a revised version of the manuscript that addresses the points raised during the review process. Please pay special attention to the Reviewer's 1 remark on the infinite number of trips in case of the zero infection and fix the function accordingly.

We look forward to receiving your revised manuscript.

Kind regards,

Itzhak Benenson, Ph.D.

Academic Editor

PLOS ONE

Journal Requirements:

“This work was supported  by the Research Institute for Mathematical Sciences, International Joint Usage/Research Center located at Kyoto University and the Open-type Professional Development Program (2021-SHIKOIN-7007). N.F. was supported by JSPS KAKENHI Grant Number JP18K11462 and Promoting Grants for Research Toward Resilient Society, 2021. R.I. was supported by JSPS KAKENHI Grant Number JP18K01561.”

“・Initials of the authors who received each award

→N.F. and R.I.

・Grant numbers awarded to each author

→JSPS KAKENHI Grant Number JP18K11462 and JP18K01561

・The full name of each funder

→Grants-in-Aid for Scientific Research

・URL of each funder website

→https://www.jsps.go.jp/index.html

・Did the sponsors or funders play any role in the study design, data collection and analysis, decision to publish, or preparation of the manuscript?

→NO”

5. We note that Figures 4, 5a and 5b in your submission contain map images which may be copyrighted. All PLOS content is published under the Creative Commons Attribution License (CC BY 4.0), which means that the manuscript, images, and Supporting Information files will be freely available online, and any third party is permitted to access, download, copy, distribute, and use these materials in any way, even commercially, with proper attribution. For these reasons, we cannot publish previously copyrighted maps or satellite images created using proprietary data, such as Google software (Google Maps, Street View, and Earth). For more information, see our copyright guidelines: http://journals.plos.org/plosone/s/licenses-and-copyright.

 a. You may seek permission from the original copyright holder of Figure 4, 5a and 5b to publish the content specifically under the CC BY 4.0 license. 

Reviewers' comments:

Reviewer's Responses to Questions

**Comments to the Author**

1. Is the manuscript technically sound, and do the data support the conclusions?

Reviewer #1: Yes

Reviewer #2: Yes

2. Has the statistical analysis been performed appropriately and rigorously? 

Reviewer #1: Yes

Reviewer #2: Yes

3. Have the authors made all data underlying the findings in their manuscript fully available?

Reviewer #1: Yes

Reviewer #2: Yes

4. Is the manuscript presented in an intelligible fashion and written in standard English?

Reviewer #1: Yes

Reviewer #2: Yes

5. Review Comments to the Author

Reviewer #1: The study is about how origin destination flows in Japan change in response to infection numbers and/or lockdown measures, and in how far these sensitivities are influenced by the distance between the origin/destination or by the density in the destination zone.

This is overall a nice research question, starting from a nice data set.

I do think, however, that the study needs more work before being acceptable for a general audience journal like PLOS ONE.

The main issue starts from the specification eq 1. If one sets infections (IFC) to zero, then one obtains infinitely many trips. In consequence, this is not a specification one could just copy to be used in other models.

I think what has happened is that the authors have selected a model specification that is particularly suited for the determination of elasticities. This becomes clear in the (relatively short) section on “Total effects and their sensitivity”. Elasticities are useful short-term constructs, i.e. they can answer questions such as: Infections today are at level X, now let us assume that they grow by 50%, how will mobility change. But elasticities are not useful to describe everything from before the pandemics until after the second lockdown in one equation.

To make the model useful, the authors need to move more into that direction, i.e. how elasticities change over the course of the pandemics. A first step is done in table 2. And here one sees how unstable the results are: While table 1 finds a 22% reduction in trips by the first lockdown, table 2 (which investigates the period up to and including the first lockdown separately), that reduction is now down to 16%. At the same time, the effect of each 1% more infections increases from 0.011% to 0.027%. Personally, I would try elasticities month by month, but in the end that is up to the authors.

A second issue is that the model specification is unstable even with respect to its signs. Model 1 from Table 1 makes sense, except for LD2 (= second lockdown), which, however, could as well just be set to zero. However, in Model 2 of the same Table, the coefficients of ln(ifc), ln(long), ln(ld1), ln(ld2) are now positive, implying (for example) that higher infection numbers mean more trips. This is, presumably, just there to compensate for some effect of the crossterms, but for a model that is useful in practice one should not use the lower order terms to correct for oddities in the higher order terms. (In this case, possibly the model is misspecified for small values of “distance” or “density”.)

This goes along with the fact that distance and density (and some other things) need units. It then becomes clear that, for models that use logarithms, the units are also free parameters, i.e. presumably the model should be beta_dens * ln(dens/dens_*), where dens_* should also be estimated. Or maybe even beta_dens * ln( 1 + dens/dens_*) to get rid of the singularity near zero. I am not a statistician, but, as stated, a model with as many oddities as the present model will be impossible to use in practice.

Minor:

Why are IFC and LONG not divided by the population size? By not dividing them, one pretends that having 300 infection cases in a population of 3000 is the same as having it in a population of 30’000.

Fig 5a and 5b look the same to me.

I would like to see the resulting alpha_t (or exp(alpha_t)) after the estimations where applicable (e.g. Table 1 models 3 and 4). One would expect that this will trace overall mobility quite well.

Reviewer #2: This paper extends over a rather long series of publications using cellphone data to explore the effects of measures imposed by governments to fight the COVID-19 pandemic on human mobility. It extends first by focusing explicitly on spatial characteristics of daily trips, and second, it uses a longer data series and thus is able to study potential changes in mobility behavior over time.

Like in other papers studying changes in human mobility during the COVID-19 lockdowns, the results confirm the expected -- or from the perspective of governments, intended -- changes, namely that the lockdowns have had an effect. The present manuscript uses data from Japan, where lockdowns have only been non-compulsory. The effects of two lockdowns are studied, with the first one lasting from April 7 to May 25, 2020, and the second lasting from January 8 to March 21, 2021. The effects are significant but rather weak, particularly in the second lockdown. Nevertheless, the results confirm that even a non-compulsory lockdown can have an effect on people’s mobility. In particular, this study found that long-distance trips and trips to crowded places decrease significantly when reported infections increase. Also, it showed that the second lockdown was less effective than the first.

Specific comments

1) L4, L11: “according to” not needed in both cases

2) L16-L19: The two sentences here need to be better connected. There needs to be a better transition explaining why economic reasons (and not epidemiological reasons) necessitate the study of changes in daily mobility by lockdowns.

3) L25: What is meant by “Place IQ”? do you mean PlaceIQ.com?

4) L38-L45: Actually, there have been quite a few studies using cellphone data to study changes in daily mobility during the COVID-19 pandemic, including ones that looked at spatial patterns of mobility. I suggest you do a more extensive literature search and identify more accurately the research gaps that you are responding to, and hence the novelty of the paper.

5) L52-L53: The phrase “they capture people's hourly location at 500m x 500 m cell level with their residency” is not clear. Do you mean “within their residency”? I suppose you meant to say that the cellphone data have been aggregated to grid cells of 500 m resolution, but what does the residency do here?

6) L64: “Summary of” not needed.

7) Figure 1: The line width used is too heavy, which particulary renders 1B illegible (the curve for Saitama is not visible).

8) L74: “Review of” not needed.

9) L114: After “was the same”, insert a reference to Fig. 1B.

10) L135: Is NHK the Japanese government health agency? Needs some explanation.

11) L143: “the QGIS”  “QGIS”

12) L152: “time zone” is ambiguous.

13) L163: “reasonability” sounds awkward in this context.

14) Figure 4: This figure has several cartographic problems:

- The figure-ground relationship is not clear. What is sea, what is land? For instance, using a shaded relief map in the background could clarify that.

- The figure needs a scalebar.

- The city limits of Tokyo should be shown. Same for the study area, if the study area extends over the city limits.

- Additional labelling, or adding a shaded relief map in the background, could provide better spatial reference. For non-Japanese readers, it’s hard to understand what we see here.

15) L182: Why 4000 people? Why this number?

16) Figure 5: Fig. 5A was included twice, while Fig. 5B was missing in the manuscript. However, the correct figure was contained in the file Figs.pdf that was available for download.

17) Figure 5 is too small to understand what’s happening here. I suggest adding an inset map showing a zoomed-in portion that is particularly interesting for the story of the paper.

18) L196: Change this title to “Methods”

19) L213: What does the “it” in “Third, it also includes” refer to?

20) L220:

- Before L220, insert a new section title “Results”. (Same heading hierarchy level as “Methods”)

- Change “Results of the baseline estimation” to “Baseline estimation”

21) L229: I can’t see 22.2% and 0.26% in Table 1.

22) Footnote 12: Sections are not numbered. Hence, the reference to Section 3.1 won’t work.

23) Footnote 13: change “unlike” to “except”

24) L276: change “almost negative” to “slightly negative”? “almost” seems odd.

25) Footnote 17: September 2020

6. PLOS authors have the option to publish the peer review history of their article (what does this mean?). If published, this will include your full peer review and any attached files.

Reviewer #1: No

Reviewer #2: No

---

## [Author Response · Author response to Decision Letter 0]

16 Aug 2022

Reply to the comments from Reviewer #1:

Thank you so much for your really helpful suggestions on reliability of the model’s

prediction for small variables. We modify all the estimation following them, although the

results did not change significantly. We also add more detailed estimation and discussion

for time series trend of the mobility Following your advices. The replies to each of them

are as follows:

Q1.

The main issue starts from the specification eq 1. If one sets infections (IFC) to zero, then

one obtains infinitely many trips. In consequence, this is not a specification one could just

copy to be used in other models.

A1.

Although we actually used ln(1+IFC) in the submitted paper, it was not clearly mentioned.

Therefore, the original model can be applied to the case with IFC=0. However, we

conducted more elaborate model specification, following the manner mentioned in

Q3;please see A3 for more details. Please refer to “Methodology” and “Results” in the

manuscript.

Q2.

I think what has happened is that the authors have selected a model specification that is

particularly suited for the determination of elasticities. This becomes clear in the

(relatively short) section on "Total effects and their sensitivity". Elasticities are useful

short-term constructs, i.e. they can answer questions such as:

Infections today are at level X, now let us assume that they grow by 50%, how will

mobility change. But elasticities are not useful to describe everything from before the

pandemics until after the second lockdown in one equation. To make the model useful,

the authors need to move more into that direction, i.e. how elasticities change over the

course of the pandemics. A first step is done in table 2. And here one sees how unstable

the results are: While table 1 finds a 22% reduction in trips by the first lockdown, table 2

(which investigates the period up to and including the first lockdown separately), that

reduction is now down to 16%. At the same time, the effect of each 1% more infections

increases from 0.011% to 0.027%. Personally, I would try elasticities month by month,

but in the end that is up to the authors.

A2.

We agree with your comment. In reality, elasticity of mobility regarding new infection

cases, (dP/dIFC)/(P/IFC), will differ depending on time and situation because of the

following two reasons:

1. Sensitivity or coefficient ¥beta might change, although our basic analysis assumes

that it is fixed throughout the data period.

2. Elasticity of X (e.g. number of new infection cases) depends on the current value of

X if we employ the function ¥beta ln(1+X), although this problem does not matter

when we use ln (X).

Since the reviewer mentions the first problem, we specifically focused on it. We tried

separating the data periods into five segments, where the length of one segment is about

three months. However, further detailed segmentation is hard to implement because our

data point is only once a week (i.e., Thursday at 10:00 AM) to control the effect of day

and time. Please see “Trend of trip patterns” in “Results.” Figure 9 simply shows the

“total effect” in each period, which is almost the same as the elasticity if X is sufficiently

high (e.g., more than 100). Please see the “Trend of trip patterns.”

For the second problem, we needed to calculate the elasticity for different X. However,

we actually employed the function as ¥beta ln(1+X/0.5) for the number of infection cases,

where 0.5 is set to maximize the within R2 of the model following Comment 3. Since X

is much more than 100 in many cases, “1” in ln(1+X/0.5) is almost negligible. Therefore,

the difference of X does not matter so much for the value of elasticity. Please see L290-

291 and Footnote 15.

Q3.

A second issue is that the model specification is unstable even with respect to its signs.

Model 1 from Table 1 makes sense, except for LD2 (= second lockdown), which,

however, could as well just be set to zero. However, in Model 2 of the same Table, the

coefficients of ln(ifc), ln(long), ln(ld1), ln(ld2) are now positive, implying (for example)

that higher infection numbers mean more trips. This is, presumably, just there to

compensate for some effect of the crossterms, but for a model that is useful in practice

one should not use the lower order terms to correct for oddities in the higher order terms.

(In this case, possibly the model is misspecified for small values of “distance” or

“density”.)

This goes along with the fact that distance and density (and some other things) need units.

It then becomes clear that, for models that use logarithms, the units are also free

parameters, i.e. presumably the model should be beta_dens * ln(dens/dens_*), where

dens_* should also be estimated. Or maybe even beta_dens * ln( 1 + dens/dens_*) to get

rid of the singularity near zero. I am not a statistician, but, as stated, a model with as many

oddities as the present model will be impossible to use in practice.

A3. To solve the problem in the model with the cross terms, we used the functional form

ln( 1 + x/¥omega^x), where x is IFC, Dens, or Dist. The unit of each variable, denoted by

¥omega^x, is chosen to maximize the within R^2 of the model. Because of the

computational burden from our large dataset, we examined 27 patterns (3*3*3 patterns)

of ¥omega^x and chose the best one among them). Please see “Methods,” especially

L232-238.

However, even after considering such flexible specification, the problem is not

completely solved; the effect of IFC is positive for about 17% OD samples in our dataset

with short distance and low density (i.e., about 5km and 1000pop/cell). Although this

seems counterintuitive, some previous research also indicated that people have shifted to

low-density and short-distance trips, so we mentioned them to support our result. Please

see L312-315.

Further, for LD 1&2, no positive significant effect is predicted for any OD samples,

unlike IFC. This result may be explained by the “intervention effect” of compulsory

lockdown as well as the information effect, as mentioned in Watanabe and Yabu (2021).

Please see L319-321.

Minor:

Q4.

Why are IFC and LONG not divided by the population size? By not dividing them, one

pretends that having 300 infection cases in a population of 3000 is the same as having it

in a population of 30’000.

A4. Following your suggestion, we defined IFC by using the number of new infection

cases divided by relative population in order to control the difference of population size

among prefecture. The relative population size of prefecture i is defined by (population

of prefecture i)/(population of Tokyo), and the number of infections is divided by the

relative population. The absolute value of population is absorbed when we choose the

unit of population. However, such control of population is not necessary for LONG

because it is the ration of infection cases between last week and two weeks ago; hence,

population difference does not matter. Please see L214-215.

Q5.

Fig 5a and 5b look the same to me.

A5.

This is our mistake, so we replaced the figures with the correct ones.

Q6

I would like to see the resulting alpha_t (or exp(alpha_t)) after the estimations where

applicable (e.g. Table 1 models 3 and 4). One would expect that this will trace overall

mobility quite well.

A6

We check the estimates of the data fixed effects, and show them in Figure 8. The trend of

fixed effect shows how the basic mobility of people changes by period. Although the

effect of the two lockdowns are also included in them because they are not separable(so

we omit LD1&2 from Model 3,4 in table 1) as mentioned in the mainbody of the paper,

they also reveals some hidden attitudes and trend regarding mobility.

The findings from the date fixed effects are as follows. First, alpha_t gets exceptionally

low during the first lockdown because of the effect of LD1 in it.

Second, alpha_t is continuously high in the latter half of the data. In these periods, it is

considered that people are used to the pandemic.

Reply to the comments from Reviewer #2:

Thank you for your suggestion regarding related literature that makes our contribution

clearer. They were helpful for revising the Introduction. Other detailed comments on our

writing were also helpful for clarification, so we considered each comment carefully. The

replies to each of them are as follows:

1) L4, L11: “according to” not needed in both cases

A. We omitted this phrase per your suggestion.

2) L16-L19: The two sentences here need to be better connected. There needs to be a

better transition explaining why economic reasons (and not epidemiological reasons)

necessitate the study of changes in daily mobility by lockdowns.

A. We added sentences to mention why we focused on compulsory lockdowns and

people’s “behavior” from the viewpoint of economic damage. Please see the second

paragraph of the Introduction. (L17-21)

3) L25: What is meant by “Place IQ”? do you mean PlaceIQ.com?

A. We omitted the original sentence which included the term when we revised the second

paragraph of the Introduction.

4)L38-L45: Actually, there have been quite a few studies using cellphone data to study

changes in daily mobility during the COVID-19 pandemic, including ones that looked at

spatial patterns of mobility. I suggest you do a more extensive literature search and

identify more accurately the research gaps that you are responding to, and hence the

novelty of the paper.

A. We added and referred to several previous studies that use cellphone data. Although

mobile phone data has been used by a growing number of recent studies, not as many

studies focus on decision-making regarding trips during the COVID-19 pandemic; hence,

our contribution is to add new results to the literature. Please see the third paragraph of

the Introduction around L18-33.

5) L52-L53: The phrase “they capture people’s hourly location at 500m x

500 m cell level with their residency” is not clear. Do you mean “within

their residency”? I suppose you meant to say that the cellphone data

have been aggregated to grid cells of 500 m resolution, but what does

the residency do here?

A. We changed the sentence as: “Our data, collected and estimated from cellphones, are

suitable for investigating intra-city trips because they capture how many people who

have residency in each city are within a 500 x500 m cell each hour.”(L58-59) Since

the definition of the data may be difficult to understand for most readers, we added a

new figure (Fig. 2) to show the structure of the dataset.

6) L64: “Summary of” not needed.

A. We omitted these words.

7) Figure 1: The line width used is too heavy, which particularly renders

1B illegible (the curve for Saitama is not visible).

A. We made the lines thinner, though identifying the differences among the prefectures

in the early stage of the pandemic is still difficult due to the scale of the figure.

8) L74: "Review of" not needed.

A. We omitted the words.

9) L114: After "was the same", insert a reference to Fig. 1B.

A. We revised the sentence as “…the period of quasi-lockdown was the same as shown

in Fig. 1B.” (L121)

10) L135: Is NHK the Japanese government health agency? Needs some explanation.

A. NHK is an acronym for Nippon Hoso Kyokai” (Japan Broadcasting Corporation),

which is the public broadcaster in Japan. Please see L143

11) L143: "the QGIS"  "QGIS"

A. We revised this according to your suggestion.

12) L152: "time zone" is ambiguous.

A. We changed the sentence to “Although the objectives of trips are not available in our

data, we are especially interested in commuting trips, and measuring the temporary

population at 10:00 AM is appropriate because most people finish commuting by

then.”(L160-161)

13) L163: "reasonability" sounds awkward in this context.

A. We revised as follows: “To check the validity of this idea, we also show the locations

at 9:00 and 11:00.”(L171)

14) Figure 4: This figure has several cartographic problems:

- The figure-ground relationship is not clear. What is sea, what is

land? For instance, using a shaded relief map in the background could clarify that.

- The figure needs a scalebar.

- The city limits of Tokyo should be shown. Same for the study area, if the study area

extends over the city limits.

- Additional labelling, or adding a shaded relief map in the background, could provide

better spatial reference. For non-Japanese readers, it's hard to understand what we see

here.

A. We revised Figure 5(number of the figure changes) to show the geography of Japan

and the targeted area clearly.

15) L182: Why 4000 people? Why this number?

A. The number (4000 people) is obtained from looking at Figure4, and we never use 4000

for a threshold in the later analysis except for Figure5. So we revised the sentence as: “Fig

5 shows that change in mobility occurs in grid cells with high population density, which

seems to be especially significant for a populations larger than 4000.”(L188-190)

16) Figure 5: Fig. 5A was included twice, while Fig. 5B was missing in

the manuscript. However, the correct figure was contained in the file

Figs.pdf that was available for download.

A. This is our mistake, and we corrected Figures 6 a,b. (number of the figure changes).

17) Figure 5 is too small to understand what's happening here. I suggest

adding an inset map showing a zoomed-in portion that is particularly

interesting for the story of the paper.

A. We presented a zoomed in map to show what happens around the central area of Tokyo

MA. Figure 6 clearly shows that lockdown significantly affects the center rather than the

subcenter; hence, the influence of distance and density might be anticipated in a very

indirect manner, though we need to show clearer and more direct evidence. Please also

see L200-205, which mentions how this figure relates to our story.

18) L196: Change this title to "Methods"

A. We changed in this way. The new table of contents is as follows (the new sections are

colored in blue):

⚫ Introduction

⚫ Background

➢ COVID-19 in Tokyo

➢ Lockdowns in Japan and other countries

⚫ Data

➢ Data

Mobile spatial statistics data

The number of COVID-19 cases

Geographical data

The data period and targeted area

➢ Summary of the mobility data

Patterns of morning trips

The effect of quasi lockdowns

⚫ Methods

⚫ Results

➢ Baseline estimation

➢ Total effects and their sensitivity

➢ Trend of trip patterns

➢ Mobility in weekend afternoon

⚫ Conclusion

19) L213: What does the "it" in "Third, it also includes" refer to?

A. “it” means variable X_{ij,t}. Hence, we clearly mention that in this sentence. (L225)

20) L220:

- Before L220, insert a new section title "Results". (Same heading

hierarchy level as "Methods")

- Change "Results of the baseline estimation" to "Baseline estimation"

A. We revised the titles of the sections in this way. Please see the answer to Comment 18

above.

21) L229: I can't see 22.2% and 0.26% in Table 1.

A. We changed these numbers as we changed the estimation results. The revised numbers

are 21.92 and 0.28 as in Table 1. Please see L250.

22) Footnote 12: Sections are not numbered. Hence, the reference to

Section 3.1 won't work.

A. Instead of “Section 3.1,” we stated, “in the section titled Total effects and sensitivity.”

Please see Footnote 13.

23) Footnote 13: change "unlike" to "except"

A. Since the original sentence was unclear, we changed it to: “Note that LDs are dummies

and hence take the values 1 or 0, while IFC is a continuous variable whose natural

logarithm is used for the independent variable.” Please see Footnote 16.

24) L276: change "almost negative" to "slightly negative"? "almost"

seems odd.

A. We revised this sentence as: “They show that the total effect of IFC is negative in most

cases in Figure 5, except for a case with small population density.”(L299)

25) Footnote 17: September 2020

A. This footnote was omitted in the revised paper.

---

## [Decision Letter · Decision Letter 1]

26 Sep 2022

PONE-D-21-40257R1Influence of trip distance and population density on intra-city mobility patterns in Tokyo during COVID-19 pandemicPLOS ONE

Dear Dr. Tsuboi,

Thank you for submitting your manuscript to PLOS ONE. After careful consideration, we feel that it has merit but does not fully meet PLOS ONE’s publication criteria as it currently stands. Therefore, we invite you to submit a revised version of the manuscript that addresses the points raised during the review process.

We look forward to receiving your revised manuscript.

Kind regards,

Itzhak Benenson, Ph.D.

Academic Editor

PLOS ONE

Journal Requirements:

Reviewers' comments:

Reviewer's Responses to Questions

**Comments to the Author**

1. If the authors have adequately addressed your comments raised in a previous round of review and you feel that this manuscript is now acceptable for publication, you may indicate that here to bypass the “Comments to the Author” section, enter your conflict of interest statement in the “Confidential to Editor” section, and submit your "Accept" recommendation.

Reviewer #1: All comments have been addressed

2. Is the manuscript technically sound, and do the data support the conclusions?

Reviewer #1: Yes

3. Has the statistical analysis been performed appropriately and rigorously? 

Reviewer #1: Yes

4. Have the authors made all data underlying the findings in their manuscript fully available?

Reviewer #1: Yes

5. Is the manuscript presented in an intelligible fashion and written in standard English?

Reviewer #1: Yes

6. Review Comments to the Author

Reviewer #1: All my major concerns have been addressed, thank you very much for developing your paper further. Small things:

* p6, line 183: An overleaf URL has sneaked in, presumably by erroneous copy/paste.

* The Fig.10 that I have in the submission is not consistent with the description: I think that this duplicates Fig.8(B) instead of showing something about the separate periods.

I take it that it is in the own interest of the authors to address these so I do not need to see the paper again.

7. PLOS authors have the option to publish the peer review history of their article (what does this mean?). If published, this will include your full peer review and any attached files.

Reviewer #1: No

---

## [Author Response · Author response to Decision Letter 1]

10 Oct 2022

Dear Dr. Itzhak Benenson,

Thank you for sending the helpful comments. Considering them carefully, we revised the paper. The revised parts of the manuscript are indicated by blue text in the revised manuscript with track changes.

Reply to the comments from Academic Editor

Q. Please review your reference list to ensure that it is complete and correct. If you have cited papers that have been retracted, please include the rationale for doing so in the manuscript text, or remove these references and replace them with relevant current references. Any changes to the reference list should be mentioned in the rebuttal letter that accompanies your revised manuscript. If you need to cite a retracted article, indicate the article's retracted status in the References list and also include a citation and full reference for the retraction notice.

A. References were revised following Vancouver style as indicated in the Submission Guidelines: https://journals.plos.org/plosone/s/submission-guidelines. We confirmed that no papers were retracted.

Reply to the comments from Reviewer

Q1, p6, line 183: An overleaf URL has sneaked in, presumably by erroneous copy/paste.

A. Thank you so much for your comments. The following URL was removed from the revised manuscript: https://ja.overleaf.com/project/6138170906460e123b1296d5. We made sure there were no mistakes elsewhere.

Q2, The Fig.10 that I have in the submission is not consistent with the description: I think that this duplicates Fig.8(B) instead of　showing　something about the separate periods.

A. 　The Fig.10 was replaced with the correct one.

----

In addition to the reviewer’s comments, we made the following additional revisions to the manuscript.

・p.8 line 229-230: The text was revised to be more accurate.

[before] "Single terms such as distance, density, and 229 lockdown dummies in few results are excluded because they are included in the fixed effects of OD and date."

[after] "Single terms of $Dens$ and $Dist$ are excluded because they are included in the fixed effects of OD. In the same reason, $LD1$ and $LD2$ are excluded when the date fixed effects are estimated."

Several other minor corrections were made.

・Periods (.) and hyphens (-) was added in several places.

・"IFC", "LONG", and "LD" fonts were unified.

・p.1 line 3:　The number of infected persons was corrected. 

[before] 400 million -> [after] 200 million

・p.7 line 227: There were two periods, so we removed one of them.

---

## [Editor Report · Decision Letter 2]

13 Oct 2022

Influence of trip distance and population density on intra-city mobility patterns in Tokyo during COVID-19 pandemic

PONE-D-21-40257R2

Dear Dr. Tsuboi,

We’re pleased to inform you that your manuscript has been judged scientifically suitable for publication and will be formally accepted for publication once it meets all outstanding technical requirements.

Kind regards,

Itzhak Benenson, Ph.D.

Academic Editor

PLOS ONE
---

## [Editor Report · Acceptance letter]

18 Oct 2022

PONE-D-21-40257R2 

Influence of trip distance and population density on intra-city mobility patterns in Tokyo during COVID-19 pandemic 

Dear Dr. Tsuboi:

I'm pleased to inform you that your manuscript has been deemed suitable for publication in PLOS ONE. Congratulations! Your manuscript is now with our production department. 

Kind regards, 

on behalf of

Professor Itzhak Benenson 

Academic Editor

PLOS ONE